# Very Deep VAEs Generalize Autoregressive Models and Can Outperform Them on Images

**Rewon Child**
OpenAI
San Francisco, CA
`rewon@openai.com`

## Abstract

We present a hierarchical VAE that, for the first time, generates samples quickly *and* outperforms the PixelCNN in log-likelihood on all natural image benchmarks. We begin by observing that, in theory, VAEs can actually represent autoregressive models, as well as faster, better models if they exist, when made sufficiently deep. Despite this, autoregressive models have historically outperformed VAEs in log-likelihood. We test if insufficient depth explains why by scaling a VAE to greater stochastic depth than previously explored and evaluating it CIFAR-10, ImageNet, and FFHQ. In comparison to the PixelCNN, these very deep VAEs achieve higher likelihoods, use fewer parameters, generate samples thousands of times faster, and are more easily applied to high-resolution images. Qualitative studies suggest this is because the VAE learns efficient hierarchical visual representations. We release our source code and models at `https://github.com/openai/vdvae`.

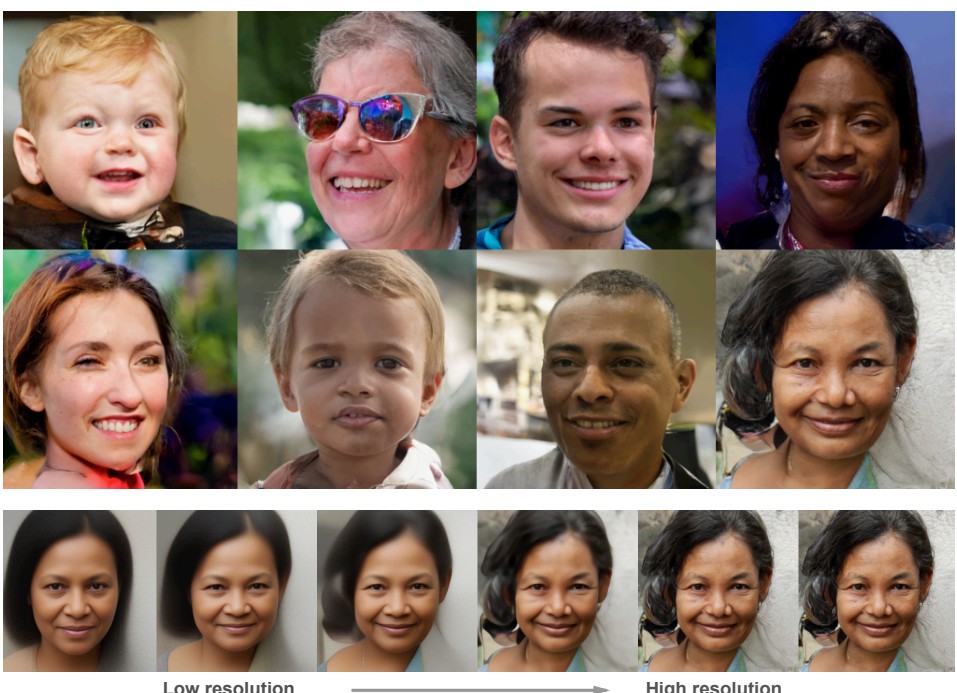

Low resolution ⟶ High resolution

Figure 1: **Selected samples from our very deep VAE on FFHQ-256, and a demonstration of the learned generative process.** VAEs can learn to first generate global features at low resolution, then fill in local details in parallel at higher resolutions. When made sufficiently deep, this learned, parallel, multiscale generative procedure attains a higher log-likelihood than the PixelCNN.

# 1 INTRODUCTION

One potential path to increased data-efficiency, generalization, and robustness of machine learning methods is to train generative models. These models can learn useful representations without human supervision by learning to create examples of the data itself. Many types of generative models have flourished in recent years, including likelihood-based generative models, which include autoregressive models (Uria et al., 2013), variational autoencoders (VAEs) (Kingma & Welling, 2014; Rezende et al., 2014), and invertible flows (Dinh et al., 2014; 2016). Their objective, the negative log-likelihood, is equivalent to the KL divergence between the data distribution and the model distribution. A wide variety of models can be compared and assessed along this criteria, which corresponds to how well they fit the data in an information-theoretic sense.

Starting with the PixelCNN (Van den Oord et al., 2016), autoregressive models have long achieved the highest log-likelihoods across many modalities, despite counterintuitive modeling assumptions. For example, although natural images are observations of latent scenes, autoregressive models learn dependencies solely between observed variables. That process can require complex function approximators that integrate long-range dependencies (Oord et al., 2016; Child et al., 2019). In contrast, VAEs and invertible flows incorporate latent variables and can thus, in principle, learn a simpler model that mirrors how images are actually generated. Despite this theoretical advantage, on the landmark ImageNet density estimation benchmark, the Gated PixelCNN still achieves higher likelihoods than all flows and VAEs, corresponding to a better fit with the data.

Is the autoregressive modeling assumption actually a better inductive bias for images, or can VAEs, sufficiently improved, outperform autoregressive models? The answer has significant practical stakes, because large, compute-intensive autoregressive models (Strubell et al., 2019) are increasingly used for a variety of applications (Oord et al., 2016; Brown et al., 2020; Dhariwal et al., 2020; Chen et al., 2020). Unlike autoregressive models, latent variable models only need to learn dependencies between latent and observed variables; such models can not only support faster synthesis and higher-dimensional data, but may also do so using smaller, less powerful architectures.

We start this work with a simple but (to the best of our knowledge) unstated observation: hierarchical VAEs *should* be able to at least match autoregressive models, because autoregressive models are equivalent to VAEs with a powerful prior and restricted approximate posterior (which merely outputs observed variables). In the worst case, VAEs should be able to replicate the functionality of autoregressive models; in the best case, they should be able to learn better latent representations, possibly with much fewer layers, if such representations exist.

We formalize this observation in Section 3, showing it is only true for VAEs with more stochastic layers than previous work has explored. Then we experimentally test it on competitive natural image benchmarks. Our contributions are the following:

- We provide theoretical justification for why greater depth (up to the data dimension $D$, but also as low as some value $K \ll D$) could improve VAE performance (Section 3)

- We introduce an architecture capable of scaling past 70 layers, when previous work explored at most 30 (Section 4)

- We verify that depth, independent of model capacity, improves log-likelihood, and allows VAEs to outperform the PixelCNN on all benchmarks (Section 5.1)

- Compared to the PixelCNN, we show the model also uses fewer parameters, generates samples thousands of times more quickly, and can be scaled to larger images. We show evidence these qualities may emerge from the model learning an efficient hierarchical representation of images (Section 5.2)

- We release code and models at `https://github.com/openai/vdvae`.

# 2 PRELIMINARIES

We review prior work and introduce some of the basic terminology used in the field.

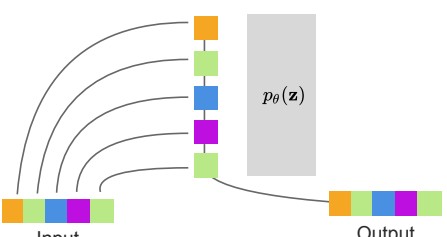
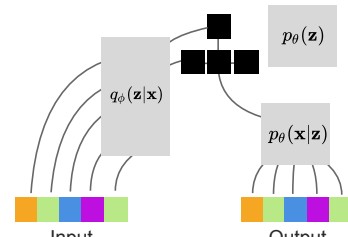

Figure 2: **Different possible learned generative models in a VAE. Left:** A hierarchical VAE can learn an autoregressive model by using the deterministic identity function as an encoder, and learning the autoregression in the prior. **Right:** Learning the encoder can lead to efficient hierarchies of latent variables (black). If the bottom group of three latent variables is conditionally independent given the first, they can be generated in parallel within a single layer, potentially leading to faster sampling.

## 2.1 VARIATIONAL AUTOENCODERS

Variational autoencoders (Kingma & Welling, 2014; Rezende et al., 2014) consist of a *generator* $p_\theta(\boldsymbol{x}|\boldsymbol{z})$, a *prior* $p_\theta(\boldsymbol{z})$, and an *approximate posterior* $q_\phi(\mathbf{z}|\mathbf{x})$. Neural networks $\phi$ and $\theta$ are trained end-to-end with backpropagation and the reparameterization trick in order to maximize the evidence lower bound (ELBO):

$$\log p_\theta(\mathbf{x}) \geq E_{\mathbf{z} \sim q_\phi(\mathbf{z}|\mathbf{x})} \log p_\theta(\mathbf{x}|\mathbf{z}) - D_{KL}[q_\phi(\mathbf{z}|\mathbf{x})||p_\theta(\mathbf{z})] \tag{1}$$

See Kingma & Welling (2019) for an in-depth introduction. There are many choices for what networks are used for $p_\theta(\boldsymbol{x}|\boldsymbol{z})$, $q_\phi(\boldsymbol{z}|\boldsymbol{x})$, and whether $p_\theta(\boldsymbol{z})$ is also learned or set to a simple distribution.

We study VAEs with independent $p_\theta(\boldsymbol{x}|\boldsymbol{z})$ – that is, where each observed $x_i$ is output without conditioning on any other $x_j$. This ensures generation time does not increase linearly with the dimensionality of the data, and requires that these VAEs learn to incorporate the complexity of the data into a rich distribution over latent variables $\boldsymbol{z}$. It is possible to have autoregressive $p_\theta(\boldsymbol{x}|\boldsymbol{z})$ (Gulrajani et al., 2016), but generation is slow for these models. They also sometimes ignore latent variables entirely, becoming equivalent to normal autoregressive models (Chen et al. (2016)).

## 2.2 HIERARCHICAL VARIATIONAL AUTOENCODERS

Much of the early work on VAEs incorporate fully-factorized Gaussian $q_\phi(\boldsymbol{z}|\boldsymbol{x})$ and $p_\theta(\boldsymbol{z})$. This can lead to poor outcomes if the latent variables required for good generation take on a more complex distribution, as is common with independent $p_\theta(\boldsymbol{x}|\boldsymbol{z})$. One of the simplest methods of gaining greater expressivity in both distributions is to use a hierarchical VAE, which has several *stochastic layers* of latent variables. These variables are emitted in groups $\boldsymbol{z}_0, \boldsymbol{z}_1, ..., \boldsymbol{z}_N$, which are conditionally dependent upon each other in some way. For images, latent variables are typically output in feature maps of varying resolutions, with $\boldsymbol{z}_0$ corresponding to a small number of latent variables at low resolution at the "top" of the network, and $\boldsymbol{z}_N$ corresponding to a larger number of latent variables at high resolution at the "bottom".

One particularly elegant conditioning structure is the *top-down VAE*, introduced in Sønderby et al. (2016). In this model, both the prior and the approximate posterior generate latent variables in the same order:

$$p_\theta(\boldsymbol{z}) = p_\theta(\boldsymbol{z}_0)p_\theta(\boldsymbol{z}_1|\boldsymbol{z}_0)...p_\theta(\boldsymbol{z}_N|\boldsymbol{z}_{<N}) \tag{2}$$

$$q_\phi(\boldsymbol{z}|\boldsymbol{x}) = q_\phi(\boldsymbol{z}_0|\boldsymbol{x})q_\phi(\boldsymbol{z}_1|\boldsymbol{z}_0, \boldsymbol{x})...q_\phi(\boldsymbol{z}_N|\boldsymbol{z}_{<N}, \boldsymbol{x}) \tag{3}$$

A diagram of this process appears in Figure 3. A typical implementation of this model has $\phi$ first perform a deterministic "bottom-up" pass on the data to generate features, then processes the groups

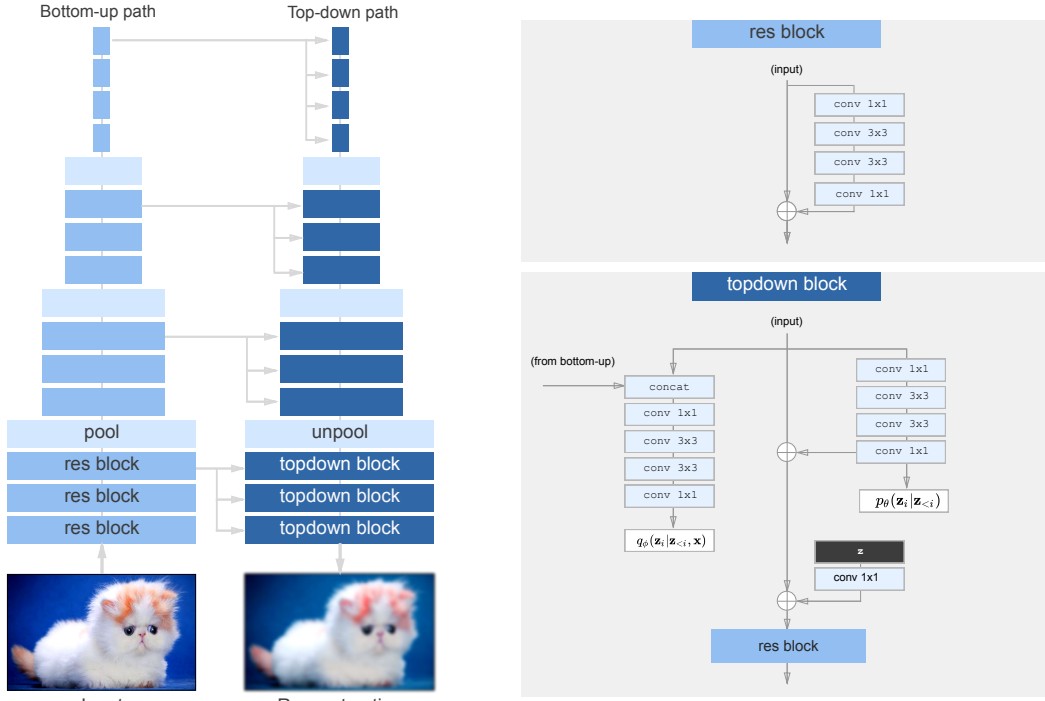

Figure 3: **A diagram of our top-down VAE architecture.** Residual blocks are similar to bottleneck ResNet blocks (He et al., 2016). Each convolution is preceded by the GELU nonlinearity (Hendrycks & Gimpel, 2016). $q_\phi(.)$ and $p_\theta(.)$ are diagonal Gaussian distributions. $\mathbf{z}$ is sampled from $q_\phi(.)$ during training, and $p_\theta(.)$ when sampling. We use average pooling and nearest-neighbor upsampling for pool and unpool layers.

of latent variables from top to bottom, using feedforward networks to generate features which are shared between the approximate posterior, prior, and reconstruction network $p_\theta(\boldsymbol{x}|\boldsymbol{z})$. We adopt this base architecture as it is simple, empirically effective, and has been postulated to resemble biological processes of perception (Dayan et al., 1995).

## 3    WHY DEPTH MATTERS FOR HIERARCHICAL VAES

We find that hierarchical VAEs with sufficient depth can not only learn arbitrary orderings over observed variables, but also learn more effective latent variable distributions, if such distributions exist. We present these results below.

**Definition** ($N$-layer VAE). A deep hierarchical VAE with $N$ stochastic layers, independent $p(\boldsymbol{x}|\boldsymbol{z})$, and the top-down factorization of the prior and approximate posterior in Equations 2-3.

**Proposition 1.** *$N$-layer VAEs generalize autoregressive models when $N$ is the data dimension*

**Proposition 2.** *$N$-layer VAEs are universal approximators of $N$-dimensional latent densities*

Proposition 1 (proof in Appendix, also visualized in Figure 2, left) leads to a possible explanation of why autoregressive models to date have outperformed VAEs: *they are deeper*, in the sense of statistical dependence. A VAE must be as deep as the data dimension $D$ (3072 layers in the case of 32x32 images) if the images truly require $D$ steps to generate.

Luckily, however, Proposition 2 (proof and further technical requirements in Appendix) suggests that shorter procedures, if they exist, are also learnable. $N = D$ is an extreme case, where the most effective latent variables $\boldsymbol{z} \in \mathbb{R}^D$ may simply be copies of the observed variables. But if for some $K < D$ there exist latent variables $\boldsymbol{z} \in \mathbb{R}^K$ that the generator can use to more efficiently compress the data, Proposition 2 states a $K$-layer VAE can learn the posterior and prior distribution over those variables.

Such shorter generative paths could emerge in two ways. First, as depicted in Figure 2 (right), if the model discovers that certain variables are conditionally independent given others, the model can generate them in parallel inside a single layer, where $q_\phi(\boldsymbol{z}_N | \boldsymbol{z}_{<N}, \boldsymbol{x}) = \prod_d q_\phi(z_N^{(d)} | \boldsymbol{z}_{<N}, \boldsymbol{x})$. We hypothesize these efficient hierarchies should emerge in images, as they contain many spatially independent textures, and study this in Section 5.2. Second, the model could learn a low-dimensional representation of the data. Dai & Wipf (2019) recently showed that when a VAE is trained on data distributed on a $K$-dimensional manifold embedded in $\mathbb{R}^D$, a VAE will only activate $K$ dimensions in its latent space, meaning that the VAE will require fewer layers unless the manifold dimension is $D$, which is unlikely to be the case for images.

It is difficult to ascertain the lowest possible value of $K$ for a given dataset, but it may be deeper than most hierarchical VAEs to date. Images have many thousands of observed variables, but early hierarchical VAEs did not exceed 3 layers, until Maaløe et al. (2019) investigated a Gaussian VAE with 15 layers and found it displayed impressive performance along a variety of measures. Kingma et al. (2016) and Vahdat & Kautz (2020) additionally explored networks up to 12 and 30 layers. (These additionally incorporated additional statistical dependencies in the approximate posterior through the usage of inverse autoregressive flow (Kingma et al., 2016), an alternative approach which we contrast with our approach in Section A.4). Nevertheless, given these results we hypothesize that greater depth may improve the performance of VAEs. In the next section, we introduce an architecture capable of scaling to a greater number of stochastic layers. In Section 5.1 we show depth indeed improves performance.

# 4    AN ARCHITECTURE FOR VERY DEEP VAES

We consider a "very deep" VAE to simply be one with greater depth than has previously been explored (and do not define it to be a specific number of layers). As existing implementations of VAEs did not support many more stochastic layers than they were trained on, we reimplemented a minimal VAE with the sole aim of increasing the number of stochastic layers. This VAE consists only of convolutions, nonlinearities, and Gaussian stochastic layers. It does not exhibit posterior collapse even for large numbers of stochastic layers. We describe key architectural choices here and refer readers to our source code for more details.

## 4.1    ARCHITECTURAL COMPONENTS AND INITIALIZATION

A diagram of our network appears in Figure 3. It resembles the ResNet VAE in Kingma et al. (2016), but with bottleneck residual blocks. For each stochastic layer, the prior and posterior are diagonal Gaussian distributions, as used in prior work (Maaløe et al., 2019).

As an alternative to weight normalization and data-dependent initialization (Salimans & Kingma, 2016), we adopt the default PyTorch weight intialization. The one exception is the final convolutional layer in each residual bottleneck block, which we scale by $\frac{1}{\sqrt{N}}$, where N is the depth (similar to Radford et al. (2019); Child et al. (2019); Zhang et al. (2019)). This residual scaling improves stability and performance with many layers, as we show in the Appendix (Table 3).

Additionally, we use nearest-neighbor upsampling for our "unpool" layer, which when paired with our ResNet architecture, allows us to completely remove the "free bits" and KL "warming up" terms that appear in related work. As we detail in the Appendix (Figure 5), when upsampling is done through transposed convolutional layer, the network may ignore layers at low resolution (for instance, 1x1 or 4x4 layers). We found no evidence of posterior collapse in any networks trained with nearest neighbor interpolation.

## 4.2    STABILIZING TRAINING WITH GRADIENT SKIPPING

VAEs have notorious "optimization difficulties," which are not frequently discussed in the literature but nevertheless well-known by practitioners. These manifest as extremely high reconstruction or KL losses and corresponding large gradient norms (up to $1e15$). We address this by skipping updates with a gradient norm above a certain threshold, set by hyperparameter. Though we select high thresholds that affect fewer than 0.01% of updates, this technique almost entirely eliminates divergence, and allows networks to train smoothly. We plot the evolution of grad norms and the

Table 1: **Loss by network with different configurations of stochastic layers on ImageNet-32** (similar trends appear on CIFAR-10). **Left**: Networks with equal number of layers, but with lower stochastic depth as described in Section 5.1. Increasing depth up to 48 layers still shows gains, which is farther than previous work has explored. **Right**: Networks with 48 layers, but distributed at different resolutions. We find higher resolutions benefit more from layers.

| Depth | Params | Test Loss | | Distribution of 48 layers | | | | | Test Loss |
|---|---|---|---|---|---|---|---|---|---|
| | | | | 32x32 | 16x16 | 8x8 | 4x4 | 1x1 | |
| 3 | 41M | 4.30 | | | | | | | |
| 6 | 41M | 4.18 | | 10 | 10 | 10 | 10 | 8 | 3.98 |
| 12 | 41M | 4.06 | | 12 | 12 | 10 | 8 | 6 | 3.97 |
| 24 | 41M | 3.98 | | 14 | 14 | 10 | 6 | 4 | 3.96 |
| 48 | 41M | 3.95 | | 16 | 16 | 10 | 4 | 2 | 3.95 |

values we select in (Figure 6). An alternative approach to stabilizing networks may be the spectral regularization method introduced in Vahdat & Kautz (2020).

# 5 EXPERIMENTS

We trained very deep VAEs on challenging natural image datasets. All hyperparameters for experiments are available in the Appendix and in our source code.

## 5.1 STATISTICAL DEPTH, INDEPENDENT OF CAPACITY, IMPROVES PERFORMANCE

We first tested whether greater statistical depth, independent of other factors, can result in improved performance. We trained a network with 48 layers for 600k steps on ImageNet-32, grouping layers to output variables independently instead of conditioning on each other. If the input for the $i$th topdown block is $x_i$, we can make $K$ consecutive blocks independent by setting $x_{i+1}, ..., x_{i+K}$ all equal to $x_i$. (Normally, $x_{i+1} = x_i + f(\text{block}(x_i))$). This technique reduces the stochastic depth without affecting parameter count. Stochastic depth shows a clear correlation with performance, even up to 48 layers, which is past what previous work has explored (Table 1, left).

We then tested our hypothesis at scale. We trained networks on CIFAR-10, ImageNet-32, and ImageNet-64 with greater numbers of stochastic layers, but with *fewer* parameters than related work (see Table 2). On CIFAR-10, we trained a model with 45 stochastic layers and only 39M parameters, and found it achieved a test log-likelihood of 2.87 bits per dim (average of 4 seeds). On ImageNet-32 and ImageNet-64, we trained networks with 78 and 75 stochastic layers and only approximately 120M parameters, and achieved likelihoods of 3.80 and 3.52.

On all tasks, these results outperform all GatedPixelCNN/PixelCNN++ models, and all non-autoregressive models, while using similar or fewer parameters. These results support our hypothesis that *stochastic depth*, as opposed to other factors, explains the gap between VAEs and autoregressive models.

## 5.2 VERY DEEP VAEs LEARN AN EFFICIENT HIERARCHICAL ORDERING

One question that emerges from the analysis in Section 3 is whether VAEs need to be as deep as autoregressive models, or whether they can learn a latent hierarchy of conditionally independent variables which are able to be synthesized in parallel. We qualitatively show this is true in Figure 4. For FFHQ-256 images, the first several layers at low resolution almost wholly determine the global features of the image, even though they only account for less than 1% of the latent variables. The rest of the high-resolution variables appear to be spatially independent, meaning they can be emitted in parallel in a number of layers much lower than the dimensionality of the image. This efficient hierarchical representation may underlie the VAE's ability to achieve better log-likelihoods than the PixelCNN while simultaneously sampling thousands of times faster. This can be viewed as a *learned* parallel multiscale generation method, unlike the handcrafted approaches of Kolesnikov & Lampert (2017); Menick & Kalchbrenner (2018); Reed et al. (2017).

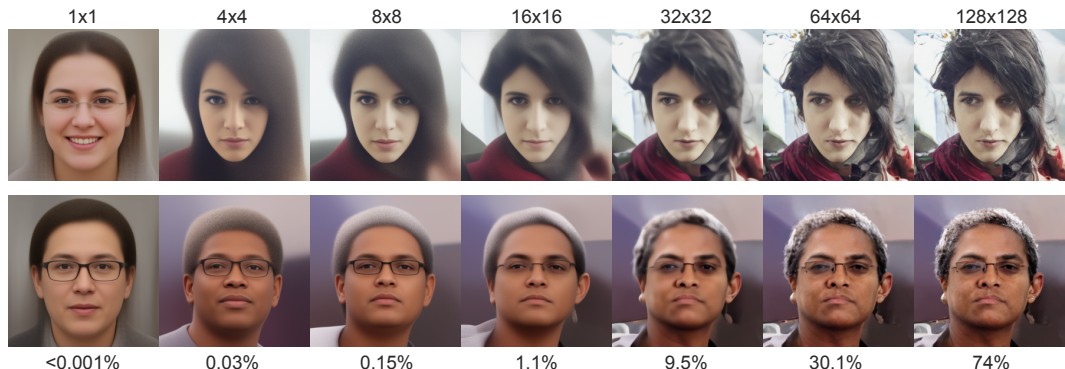

Figure 4: **Cumulative percentage of latent variables at a given resolution, and reconstructions of samples on FFHQ-256.** We sample latent variables from the approximate posterior until the given resolution, and sample the rest from the prior at low temperature. This shows what images are likely given a subset of latent variables. Low-resolution latents comprise a small fraction of the total latents, but encode significant portions of the global structure. This suggests deep VAEs learn efficient, hierarchical representations of the data.

Additionally, we found that on all datasets we tested, very deep VAEs used roughly 30% fewer parameters than the PixelCNN (Table 2). One possible explanation is that the learned hierarchical generation procedure involves fewer long-range dependencies, or may otherwise be simpler to learn.

We found that networks in general benefited from more layers at higher resolutions (Table 1, right). This suggests that global features may account for a smaller fraction of information than local details and textures, and that it is important to have many latent variables at high resolution.

### 5.2.1 VERY DEEP VAES ARE EASILY SCALED TO HIGH DIMENSIONAL DATA

Scaling autoregressive models to higher resolutions presents several challenges. First, the sampling time and memory requirements of autoregressive models increase *linearly* with resolution. This scaling makes datasets like FFHQ-256 and FFHQ-1024 intractable for naive approaches. Although clever factorization techniques have been adopted for 256x256 images (Menick & Kalchbrenner, 2018), such factorizations may not be as effective for alternate datasets or higher-resolution images.

Our VAE, in contrast, readily scales to higher resolutions. The same network used for 32x32 images can be applied to 1024x1024 images by introducing a greater number of upsampling layers throughout the network. We found we could train an equal number of steps (1.5M) using a similar number of training resources (32 GPUs for 2.5 weeks) on both 32x32 and 1024x1024 images with few hyperparameter changes (see Appendix for hyperparameters). Samples from both models (displayed in Appendix) require a single forward pass of the model to generate, with only minor differences in runtime. An autoregressive model, on the other hand, would require a thousand times more network evaluations to sample 1024x1024 images and likely require a custom training procedure.

## 6 RELATED WORK AND DISCUSSION

Our work is inspired by previous and concurrent work in hierarchical VAEs (Sønderby et al., 2016; Maaløe et al., 2019; Vahdat & Kautz, 2020). Relative to these works, we provide some justification for why deeper networks may perform better, introduce a new architecture, and empirically demonstrate gains in log-likelihood. Many aspects of prior work are complementary with ours and could be combined. Maaløe et al. (2019), for instance, incorporates a "bottom-up" stochastic path that doubles the depth of the approximate posterior, and Vahdat & Kautz (2020) introduces a number of powerful architecture components and improved training techniques. We seek here not to introduce a significantly better method than these alternatives, but to demonstrate that depth is a key overlooked factor in most prior approaches to VAEs.

Table 2: **Our main results on standard benchmark datasets.** Very deep VAEs outperform PixelCNN-based autoregressive models with fewer parameters while maintaining fast sampling. "Depth" refers to the number of stochastic layers for hierarchical VAEs (although BIVA and IAF-based networks have additional statistical dependencies). Sampling refers to the number of network evaluations per sample, and $D$ designates the dimensionality of the data. An asterisk (*) denotes our estimate of parameters. Samples for ImageNet and CIFAR-10 are in the Appendix.

| | Model type | Params | Depth | Sampling | NLL |
|---|---|---|---|---|---|
| **CIFAR-10** | | | | | |
| PixelCNN++ (Salimans et al., 2017) | AR | 53M* | | $D$ | 2.92 |
| PixelSNAIL (Chen et al., 2017) | AR | | | $D$ | 2.85 |
| Sparse Transformer (Child et al., 2019) | AR | 59M | | $D$ | **2.80** |
| VLAE (Chen et al., 2016) | VAE | | | $D$ | $\leq 2.95$ |
| IAF-VAE (Kingma et al., 2016) | VAE | | 12 | 1 | $\leq 3.11$ |
| Flow++ (Ho et al., 2019) | Flow | 31M | | 1 | $\leq 3.08$ |
| BIVA (Maaløe et al., 2019) | VAE | 103M | 15 | 1 | $\leq 3.08$ |
| NVAE (Vahdat & Kautz, 2020) | VAE | 131M | 30 | 1 | $\leq 2.91$ |
| Very Deep VAE (**ours**) | VAE | 39M | 45 | 1 | $\leq$ **2.87** |
| | | | | | |
| **ImageNet-32** | | | | | |
| Gated PixelCNN | AR | 177M* | 10 | $D$ | 3.83 |
| Image Transformer (Parmar et al., 2018) | AR | | | $D$ | **3.77** |
| BIVA | VAE | 103M* | 15 | 1 | $\leq 3.96$ |
| NVAE | VAE | 268M | 28 | 1 | $\leq 3.92$ |
| Flow++ | Flow | 169M | | 1 | $\leq 3.86$ |
| Very Deep VAE (**ours**) | VAE | 119M | 78 | 1 | $\leq$ **3.80** |
| | | | | | |
| **ImageNet-64** | | | | | |
| Gated PixelCNN | AR | 177M* | | $D$ | 3.57 |
| SPN (Menick & Kalchbrenner, 2018) | AR | 150M | | $D$ | 3.52 |
| Sparse Transformer | AR | 152M | | $D$ | **3.44** |
| Glow (Kingma & Dhariwal, 2018) | Flow | | | 1 | 3.81 |
| Flow++ | Flow | 73M | | 1 | $\leq 3.69$ |
| Very Deep VAE (**ours**) | VAE | 125M | 75 | 1 | $\leq$ **3.52** |
| | | | | | |
| **FFHQ-256 (5 bit)** | | | | | |
| NVAE | VAE | | 36 | 1 | $\leq 0.68$ |
| Very Deep VAE (**ours**) | VAE | 115M | 62 | 1 | $\leq$ **0.61** |
| | | | | | |
| **FFHQ-1024 (8 bit)** | | | | | |
| Very Deep VAE (**ours**) | VAE | 115M | 72 | 1 | $\leq$ **2.42** |

Diffusion models can be seen as deep VAEs that, like autoregressive models, have a specific analytical posterior. Ho et al. (2020) showed that such models achieve impressive sample quality with great depth, which is in line with our observations that greater depth is helpful for VAEs. One benefit of the VAEs we outline in this work over diffusion models is that our VAEs generate samples with a single network evaluation, whereas diffusion models currently require a large number of network evaluations per sample.

Inverse autoregressive flows (IAF) are also closely related, and we discuss the differences with hierarchical models in Section A.4. The work of Zhao et al. (2017) may also appear to contradict our findings, and we discuss that work in Section A.5.

## 7 CONCLUSION

We argue deeper VAEs should perform better, introduce a deeper architecture, and show it outperforms all PixelCNN-based autoregressive models in likelihood while being more efficient. We hope this encourages work in further improving VAEs and latent variable models.

### ACKNOWLEDGMENTS

We thank Aditya Ramesh, Pranav Shyam, Johannes Otterbach, Heewoo Jun, Mark Chen, Prafulla Dhariwal, Alec Radford, Yura Burda, Bowen Baker, Raul Puri, and Ilya Sutskever for helpful discussions. We also thank the anonymous reviewers for helping improve our work.

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

Table 3: **Effects of scaling residual initialization on very deep VAEs**. We trained networks with varying depths for 80k iterations. Scaling the last layer in the residual block by $\frac{1}{\sqrt{N}}$ results in higher losses for shallower networks, but lower losses and greater stability for deeper networks. The number of updates which are skipped because the gradient norm would destabilize the network is significantly reduced with scaling.

| Depth | Without scaling | | With scaling | |
|---|---|---|---|---|
| | Loss | Skipped Updates | Loss | Skipped Updates |
| 15 | 2.50 | 13 | 2.51 | 0 |
| 30 | 2.36 | 41 | 2.38 | 1 |
| 45 | 2.31 | 48 | 2.30 | 0 |
| 60 | 2.30 | 76 | 2.29 | 1 |
| 75 | Diverged | - | 2.28 | 0 |

Casper Kaae Sønderby, Tapani Raiko, Lars Maaløe, Søren Kaae Sønderby, and Ole Winther. Ladder variational autoencoders. In *Advances in neural information processing systems*, pp. 3738–3746, 2016.

Emma Strubell, Ananya Ganesh, and Andrew McCallum. Energy and policy considerations for deep learning in nlp. *arXiv preprint arXiv:1906.02243*, 2019.

Benigno Uria, Iain Murray, and Hugo Larochelle. Rnade: The real-valued neural autoregressive density-estimator. In *Advances in Neural Information Processing Systems*, pp. 2175–2183, 2013.

Arash Vahdat and Jan Kautz. Nvae: A deep hierarchical variational autoencoder. *arXiv preprint arXiv:2007.03898*, 2020.

Aaron Van den Oord, Nal Kalchbrenner, Lasse Espeholt, Oriol Vinyals, Alex Graves, et al. Conditional image generation with pixelcnn decoders. In *Advances in neural information processing systems*, pp. 4790–4798, 2016.

Hongyi Zhang, Yann N Dauphin, and Tengyu Ma. Fixup initialization: Residual learning without normalization. *arXiv preprint arXiv:1901.09321*, 2019.

Shengjia Zhao, Jiaming Song, and Stefano Ermon. Learning hierarchical features from generative models. *arXiv preprint arXiv:1702.08396*, 2017.

# A APPENDIX

## A.1 ABLATIONS OF ARCHITECTURAL COMPONENTS

First, we visualize data that suggests upsampling layers and residual connections have an impact on posterior collapse (Figure 5). Architectural differences may explain why our VAEs do not need "free bits" or KL warmups to avoid posterior collapse.

In Table 3, we show residual initialization leads to smoother and better training of very deep VAEs. Without residual initialization, very deep VAEs encounter a high number of unstable updates and have higher losses.

In Figure 6, we show the max gradient norms experienced throughout training, and show that our skipping criterion avoids a small number of updates that would destabilize the network.

## A.2 PROPOSITION 1: N-LAYER VAES GENERALIZE AUTOREGRESSIVE MODELS WHEN N IS THE DATA DIMENSION

Proposition 1 shows that an autoregressive model with an arbitrary ordering over observed variables in $x \in \mathbb{R}^N$ is equivalent to an $N$-layer VAE with an approximate posterior that simply outputs the observed variables in the given order, and a generator that performs the identity function (see Figure 2).

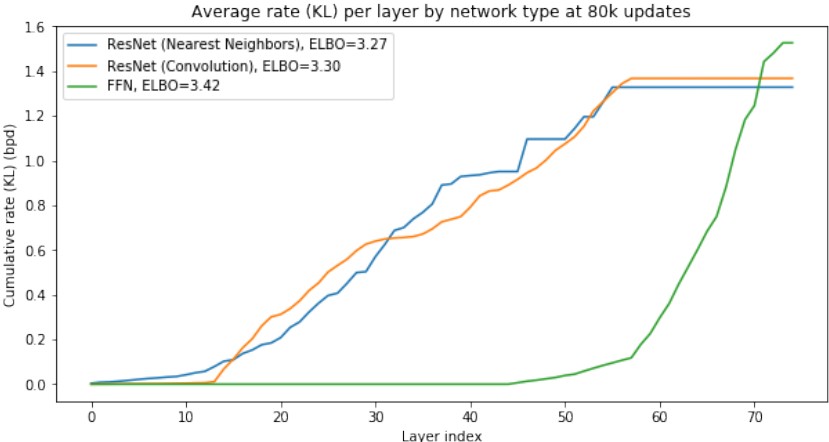

Figure 5: **Relationship between architecture and posterior collapse.** We visualize the cumulative KL divergence (or "rate", in bits per dimension) for several different architectures across a 73 layer network on ImageNet-32. When residual connections are removed from the "res block" in the top-down path (Figure 3), the model encodes no information in the first 45 layers of the network and the loss is highest ("FFN"). When a learned convolutional upsampler is used as the "unpool" layer, the first 13 layers of the network encode no information. When nearest-neighbor upsampling is used, the first layers all encode information, and the loss is the lowest.

Without loss of generality, we simplify notation by assuming each vector-valued latent variable $\boldsymbol{z}_i$ only has one element, which we write as $z_i \in \mathbb{R}$. We assume a prior and approximate posterior distribution following Equation 2 and 3.

*Proof.* Let $q(z_i = x_i | z_{<i}, \boldsymbol{x}) = 1$, and $p(x_i = z_i | \boldsymbol{z}) = 1$. Then $p(\boldsymbol{z}|\boldsymbol{x}) = q(\boldsymbol{z}|\boldsymbol{x})$, which is well-known to imply equality in the evidence lower bound (ELBO) of Eq. 1. Since $\log q(\boldsymbol{z}|\boldsymbol{x}) = \log p(\boldsymbol{x}|\boldsymbol{z}) = 0$, the ELBO becomes $\log p_\theta(\boldsymbol{x}) = \log p_\theta(\boldsymbol{z}) = \sum_{i=1}^{N} \log p_\theta(z_i | z_{<i}) = \sum_{i=1}^{N} \log p_\theta(x_i | x_{<i})$, which is equivalent to an autoregressive model over the observed variables. $\square$

### A.3 PROPOSITION 2: $N$-LAYER VAEs ARE UNIVERSAL APPROXIMATORS OF $N$-DIMENSIONAL LATENT DENSITIES

Proposition 2 shows that hierarchical VAEs learn depthwise autoregressive flows, and under certain conditions (described in Huang et al. (2017)) can express *any* density over latent variables of $N$ dimensions, given enough capacity.

*Proof.* We omit full proof, and refer readers to Huang et al. (2017); Papamakarios et al. (2019), where universality is established for autoregressive flows. Here we only note that the prior and approximate posterior in an $N$-layer VAE are autoregressive flows: Let $p_\theta(\boldsymbol{z})$ be the prior distribution. $p_\theta(\boldsymbol{z})$ can be written using the reparameterization trick as a deterministic function of noise $\epsilon$ drawn from a known base density $p_N$: $p_\theta(\boldsymbol{z}) = p_N(\epsilon) \left| \det \frac{\partial f(\epsilon, \theta)}{\partial \epsilon} \right|$, where $f$ is a neural network that implements the factorization in Eq. 2. Since $f$ is autoregressive and its Jacobian is lower triangular, $p_\theta(\boldsymbol{z})$ can approximate any $p(\boldsymbol{z})$ that fits the criteria in Huang et al. (2017). The same logic applies to $q_\phi(\boldsymbol{z}|\boldsymbol{x})$ and $p(\boldsymbol{z}|\boldsymbol{x})$. It should be noted that this result depends on $f$ being able to implement the inverse CDF of an arbitrary probability density, and so using Gaussian distributions will restrict the densities the VAE can express in practice. This is a limitation of our architecture that we nevertheless adopt since we hypothesize depth, not the elementwise density, is the more important factor. More discussion on this subject, and options for removing this restriction, are described in Huang et al. (2017) and Huang et al. (2018), and we defer studying more expressive elementwise densities to future work. $\square$

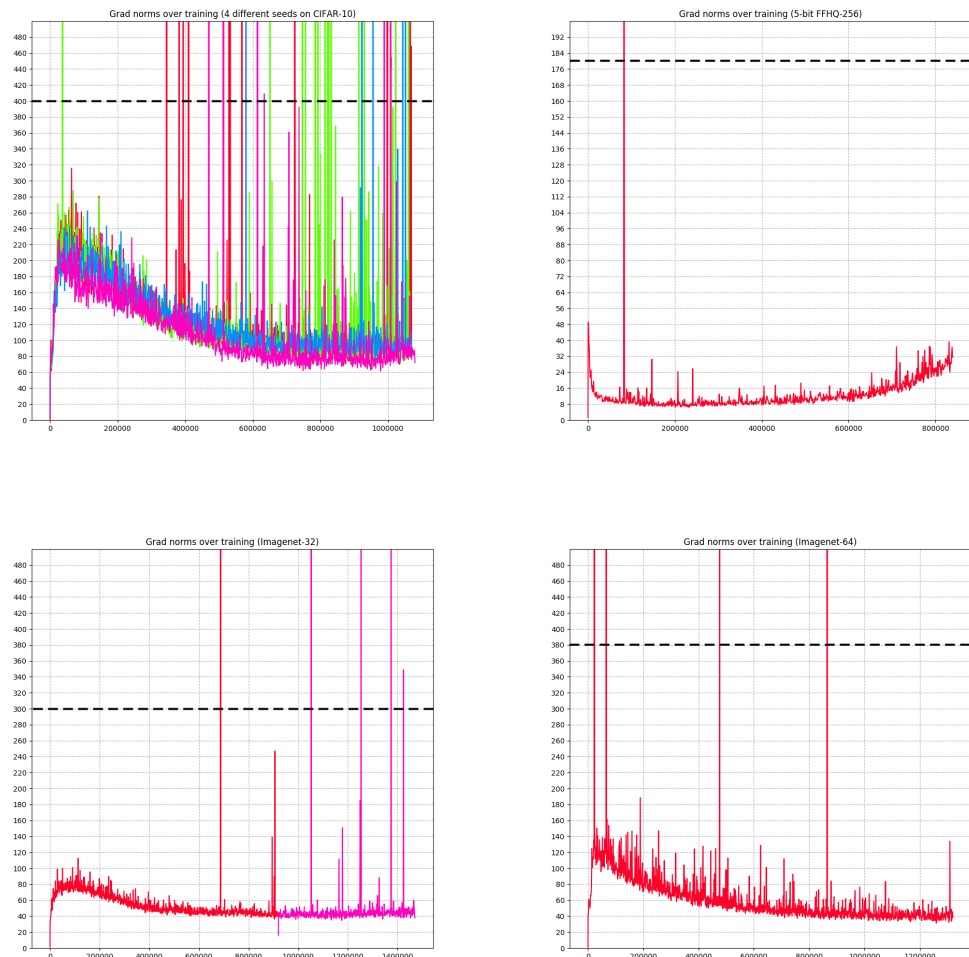

Figure 6: **Effect of gradient skipping.** We plot the max gradient norm encountered per 500 updates for our best models across datasets. The dashed black line indicates the "skip threshold", or value above which the update is skipped. We choose a high threshold that affects fewer than 0.01 percent of training updates. Without this skip heuristic, networks will diverge when extreme updates are encountered. These updates can have norm as high as $1e15$.

## A.4 A NOTE ON INVERSE AUTOREGRESSIVE FLOW

Inverse autoregressive flows (IAF, Kingma et al. (2016)) and are similar to very deep VAEs in that they are universal approximators of posterior distributions in VAEs, even with just a *single* layer and sufficiently expressive univariate density (Huang et al., 2018).

There are several practical differences between IAFs and deep hierarchical VAEs, however, which can result in qualitatively very different behavior. First, the masked autoregressive components in IAF build statistical dependencies *spatially*, whereas a very deep hierarchical VAE builds dependencies *depthwise*, and these inductive biases may better suit different domains. Additionally, IAFs spend an equal amount of computation and parameters on each variable. In contrast, a deep VAE can specify a structure, like a hierarchy of global-to-local variables, which have different computational and modeling capacities for each stage. For images, these differences may result in qualitatively different behavior, and it is not clear whether a single layer IAF can readily learn the sort of rich hierarchical decomposition of images that appear with very deep VAEs.

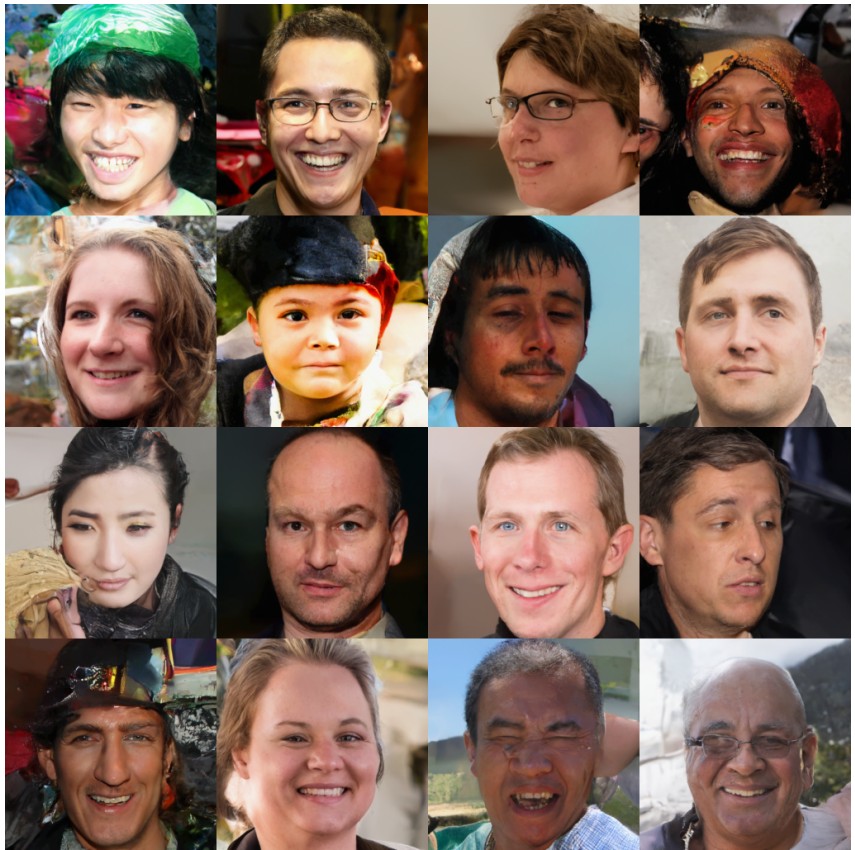

Figure 7: **Non-cherrypicked, temperature 1.0 samples on FFHQ-256.** Cover images were each cherrypicked from a batch of 16 (unadjusted temperature) samples. Here we show a random batch of 16 images for comparison.

Nevertheless, the two techniques are complementary – IAF was introduced in a deep hierarchical VAE (Kingma et al., 2016), in fact, and it is likely that introducing IAF into our architecture (as in Vahdat & Kautz (2020)) would improve performance.

### A.5    A NOTE ON LEARNING HIERARCHICAL FEATURES

The work of Zhao et al. (2017) may appear to contradict our work, by suggesting that additional layers in hierarchical VAEs *do not* lead to additional expressivity, based off their finding that Gibbs sampling from the last stochastic layer is sufficient to recover the data. For high dimensional data like images, however, the last stochastic layer may have many thousands of variables, and Gibbs sampling may take unacceptably long to converge. A hierarchy of latent variables as in our model allows efficient and tractable sampling from this distribution. Additionally, assumptions regarding global maximization of the ELBO may not apply in practice. Nevertheless, we think further clarifying these contradictory statements would be useful future work.

### A.6    BROADER IMPACT

Broadly speaking, any generative model will reflect the biases of the datasets they are trained on. If deployed without careful consideration, generative models (including but not limited to VAEs) trained on research datasets like ImageNet, CIFAR-10, and FFHQ may inadvertently cause harm by propagating or otherwise reinforcing harmful biases in the dataset. Further work is required to improve and debias research benchmark datasets to mitigate this source of negative impact.

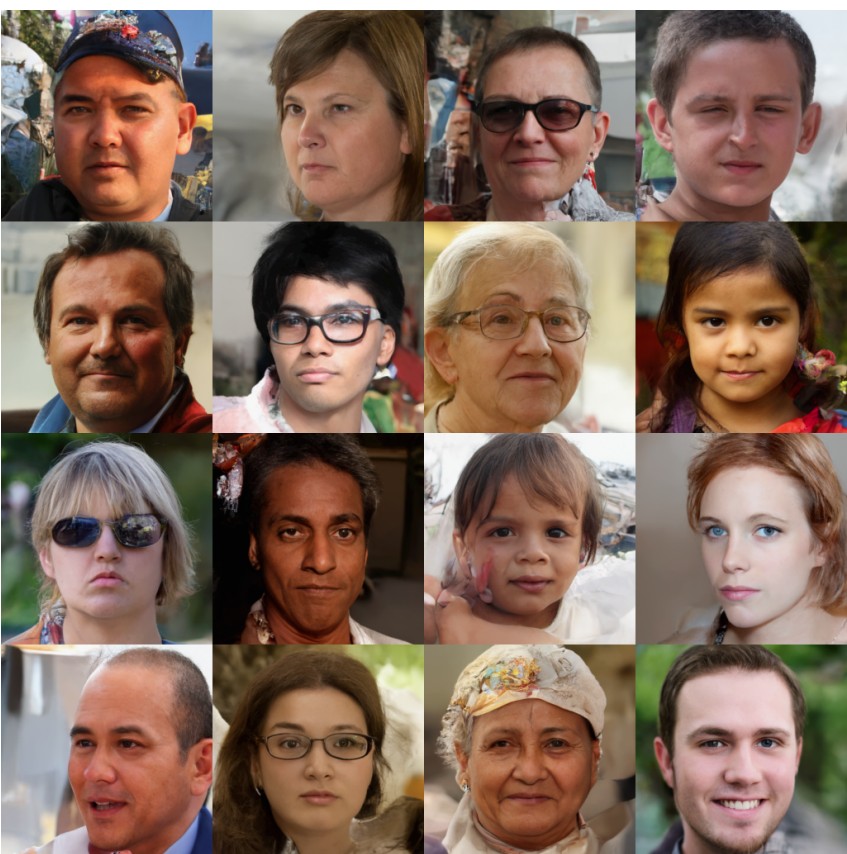

Figure 8: **Non-cherrypicked, temperature 0.85 samples on FFHQ-256.** Lower temperature samples result in greater regularity in images.

Some VAEs are distinguished from other generative models by their fast synthesis of new data examples. Generative models with fast synthesis can allow for realtime synthesis of high dimensional data, such as music, speech, and video. These models could be used to augment human creativity and lead to a number of helpful applications in real-time media applications. Such models could also be used for compression, which could assist in delivering content to bandwidth-constrained regions of the world. They can also be used for spreading disinformation, generally making it less possible to distinguish real from generated data. An additional potential harm is that fast, high quality synthesis of data could end up economically displacing individuals who rely upon creative work, such as musicians, visual artists, and more.

VAEs also are distinguished by their usage of latent variables. Generative models with useful latent variables could have positive impacts in scientific domains, where density estimation could lead to novel insights about chemical, physical, or biological data. Latent variable representations of data could also be helpful in efforts to debias, interpret, or otherwise increase understandibility of models and their representations.

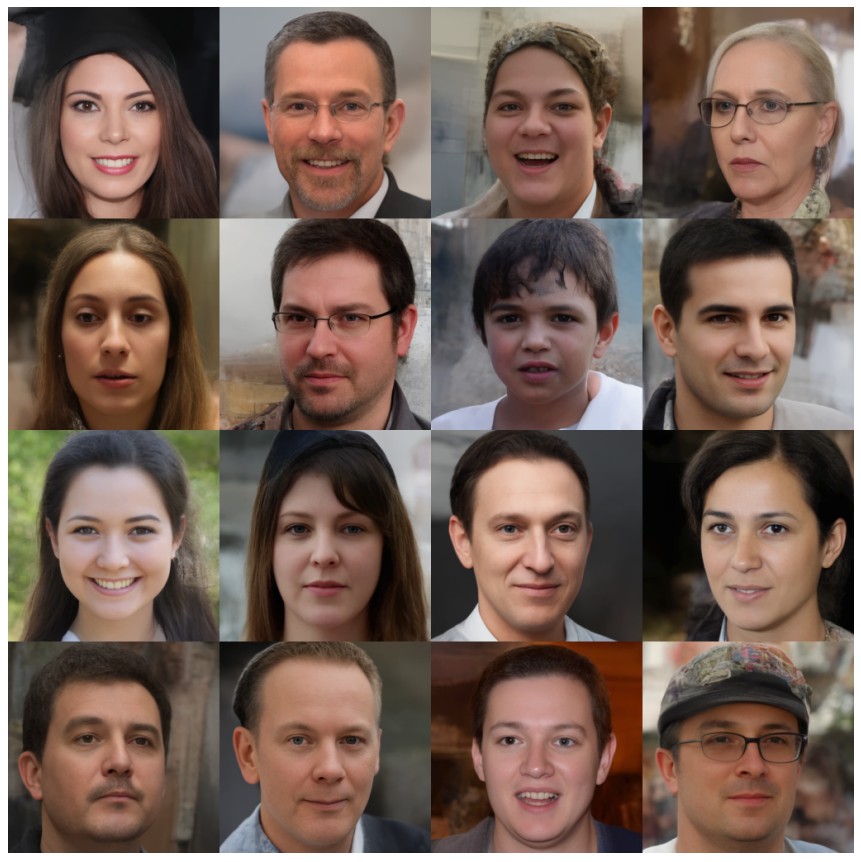

Figure 9: **Non-cherrypicked, temperature 0.60 samples on FFHQ-256.** We visualize temperature 0.60 samples for comparison with Vahdat & Kautz (2020)

Table 4: **Key hyperparameters for experiments**. We detail here the main hyperparameters used in training. FFHQ-1024 has reduced hidden size for higher resolutions; see code for details.

| Parameter | CIFAR-10 | ImageNet-32 | ImageNet-64 | FFHQ-256 | FFHQ-1024 |
|---|---|---|---|---|---|
| Num layers | 45 | 78 | 75 | 62 | 72 |
| Hidden size | 384 | 512 | 512 | 512 | Varies |
| Bottleneck size | 96 | 128 | 128 | 128 | Varies |
| Latent dim per layer | 16 | 16 | 16 | 16 | 16 |
| Batch size | 32 | 256 | 128 | 32 | 32 |
| Learning rate | 0.0002 | 0.00015 | 0.00015 | 0.00015 | 0.00007 |
| Optimizer | Adam | Adam | Adam | Adam | Adam |
| Skip threshold | 400 | 300 | 380 | 180 | 500 |
| Weight Decay | 0.01 | 0.0 | 0.0 | 0.0 | 0.0 |
| EMA rate | 0.0002 | 0.00015 | 0.00015 | 0.00015 | 0.00015 |
| Training iterations | 1.1M | 1.7M | 1.6M | 1.7M | 1.7M |
| GPUs | 2 x V100 | 32 x V100 | 32 x V100 | 32 x V100 | 32 x V100 |
| Training time | 6 days | 2.5 weeks | 2.5 weeks | 2.5 weeks | 2.5 weeks |
| Parameters | 39M | 119M | 125M | 115M | 115M |

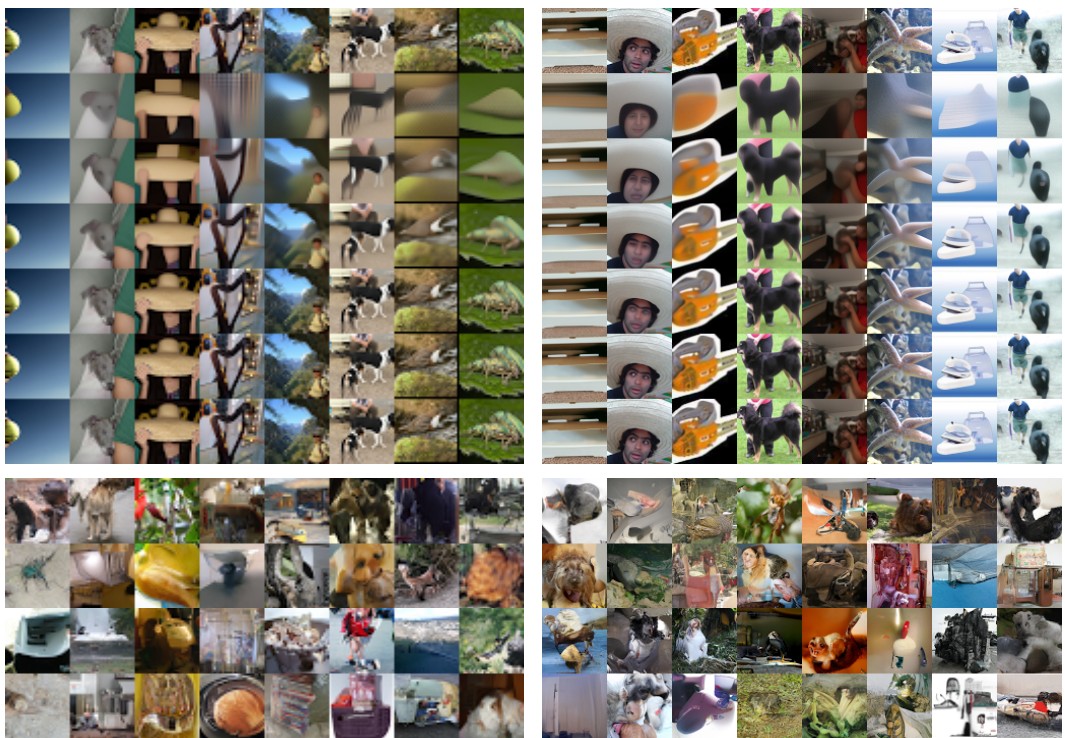

Figure 10: **ImageNet-32 (left) and ImageNet-64 (right) reconstructions and samples.** Reconstructions of validation images from various stages in the latent hierarchy (top), and unconditional samples from the model at temperature 1.0 (bottom).

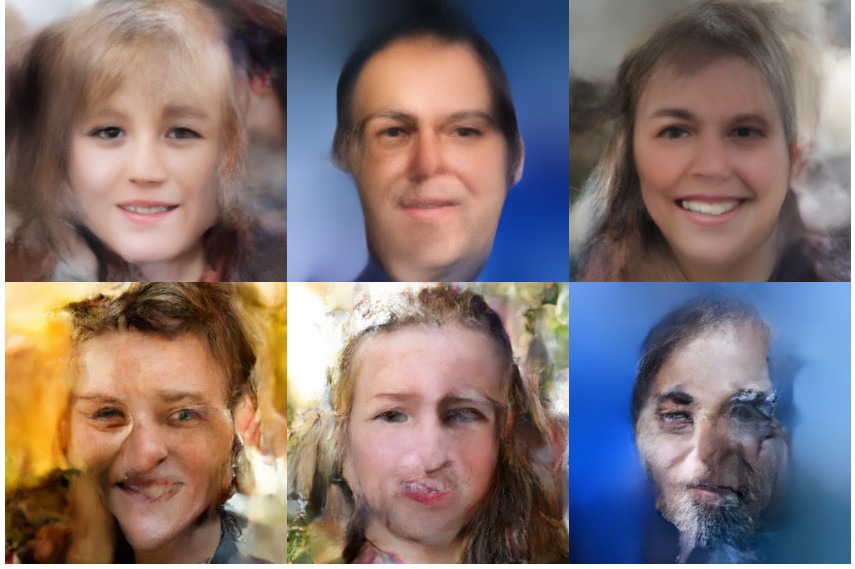

Figure 11: **FFHQ-1024 samples.** These are generated with reduced temperature (top) and temperature 1.0 (bottom). The model we train has similar capacity to smaller ones we use on 32x32, 64x64, and 256x256 images, and so fails to capture the intricacies of this more complex distribution well. A larger model, trained for longer, may achieve better sample quality.

