# OpenReview forum: "Very Deep VAEs Generalize Autoregressive Models and Can Outperform Them on Images"
_ICLR.cc/2021/Conference — ICLR 2021 Spotlight_

### Official Review · AnonReviewer4 · 2020-10-25
**A strong empirical contribution on hierarchical VAEs**

**Rating:** 7
**Confidence:** 4

**Review:**

Summary
--------------

This paper provides evidence that "very deep" hierarchical VAEs can outperform autoregressive and flow-based models albeit using less parameters on image density estimation tasks.

It seems natural to think that a hierarchy of latent variables progressively compressing information would be useful for image modelling, with top latent variables capturing more abstract/general features and bottom latent variables capturing lower-level details. However, recent success of flow based and autoregressive based models such as PixelCNN seemed to invalidate the need of such hierarchy of latent variables and to "compress" pixel-level information. Here, the authors show that a simple hierarchical VAE architecture inspired by previously proposed ones can outperform autoregressive models if it's made sufficiently "deep". I think this is an important contribution. With respect to previous work, this work relates to the concurrently proposed "Nouveau VAE" but obtains better results with less parameters and considerably less involved customization of the architecture.  The authors report impressive results on multiple datasets generally using less parameters than competing models. Additionally, sampling from very deep VAEs is considerably cheaper than in autoregressive models.

The authors also attempt at showing that learnt latent variables implement a hierarchy of information which could be useful to have in general. This point is a bit weak and not well demonstrated in the paper.

Pros
------

- Strong results on multiple tasks with a method that was previously thought to have plateaued in performance

Cons
-------

- Originality / novelty is a bit weak
- Clarity can be improved


Detailed Remarks
-------------------------

- Figure 4 is not totally convincing as high-level features in the first image are not always maintained in higher-resolution realizations (sample in the first row seem to have glasses then they disappear?). Could you include more samples to back this claim? Do you think of a way of understanding whether high-level variables maintain general info (maybe by probing the posterior samples for some downstream attribute ?)

- I find it hard to understand what is going on in Table 1 (left). In Section 5.1, referring to Table 1, what do you mean by "grouping layers to output variables independently instead of conditioning on each other" ? In Table 1, what do you mean by "masking" in the sentence "with masking introduced such that the effective stochastic depth is lower" ? I cannot find any other references to masking.

- In Figure 3, what do you use as the pooling operation? (2, 2) max pooling ?

- An ablation study of the proposed modifications to the architecture and training tricks would be useful, e.g. what's the most single important modification that makes the model work ? Is it the neighbour upsampling ? Is it the 1/\sqrt(N) init of the last layer ? Is it the skipping gradient trick ?

- How is Figure 4 obtained ? When you say "The rest of the high-resolution variables can be output in parallel, largely independent of each other", are you referring to the fact that you sample from the top 1x1 layer and then sample independently the other zs from the learnt prior e.g. p(z_4x4) ... p(z_64x64) without ancestral sampling ?

- Section A.1: "Without loss of generality, we simplify notation by assuming each vector-valued latent variable zi
only has one element, which we write as zi", do you mean each latent variable is \in R ? It'd be good to mention that
you assume an architecture with an auto-regressive learnable prior p_\theta(z_i | z_<i) or refer to Eq. 2.

- Section A.2: I am not sure you need this sentence: "Without loss of generality, we simplify notation by assuming each vector-valued latent variable zi only has one element, which we write as zi", as it seems copied from A.1.

Grammar
-------------
- Section 5.2: "an learned" -> "a learned"
- Section 4.1: "the the" -> "the"

---

> ### Author Response · Authors · 2020-11-19
> **Thank you; updated**
>
> Dear Reviewer 4,
>
> In figure 4, we now clarify in the text that it is generated by sampling from the posterior at low resolutions, then from the prior at higher resolutions. This means that the disappearance of features is not unusual; the prior distribution will generate features at high resolution that it thinks are likely, but this may be overriden by the ground truth features from the posterior. We apologize for not explaining how the diagram was created in our first draft.
>
> For Table 1, Left, we expanded the description in Section 5.1 to explain exactly how blocks of variables are emitted independently of each other. If the input for the $i$th topdown block is $x_i$, we can make $K$ consecutive blocks independent by setting $x_{i+1}$, ..., $x_{i+K-1}$ all equal to $x_i$. (Normally, $x_{i+1} = x_i + f(\mathrm{block}(x_i))$)
>
> We use average pooling and now note this in the text.
>
> At your suggestion, we incorporated an ablations table in the Appendix. Gradient skipping and residual initialization we found most important for stability at greater depths, and nearest neighbor upsampling mitigates some posterior collapse but is not strictly necessary for learning.
>
> By parallel synthesis, we mean that the network can find an arrangement where some large number of variables are conditionally independent given previous latent variables. They can then be emitted simultaneously in one layer. We revised our text to make this more clear, but please let us know if there is a phrasing that would be better.
>
> We also included all your suggested revisions for clarity. Thank you for your detailed and constructive comments.

---

### Official Review · AnonReviewer1 · 2020-10-27
**A very deep and very good VAE**

**Rating:** 8
**Confidence:** 5

**Review:**

**GENERAL**
The paper claims that high quality of generated samples and SOTA bpds are achievable by VAEs if the model is deep enough (deep in terms of the number of stochastic layers). The authors explain the architecture that resemblances the U-net architecture, and explain its building blocks. Interestingly, they are able to learn VAEs with up to 78 stochastic layers, and achieve SOTA bpds on CIFAR-10, ImageNet-32, ImageNet-64, FFHQ-256 (5-bit), and setting a great result on FFHQ-1024 (8bit).

**Strengths:**
S1: The authors are capable of training VAEs with over 45 stochastic layers (up to 78).

S2: The proposed architecture does not contain any extra "tricks", it is relatively simple. This is a great plus for the paper!

S3: The presented theorems are interesting additions to this rather practical paper.

S4 The experiments are well performed and the ablation studies are insightful.

S5: Generated images are of very high quality! Even a reflection in a glass of a generted lady is better than samples of CIFAR10 in many papers.

**Deficiencies:**
D1: The prior is not explained in the paper! Without this information, it is hard to properly understand what kind of problems occur during training q(z|x) and p(z).

**Remarks:**
R1: The proposed heuristic method for training q(z|x) reminds of the following paper:
He, J., Spokoyny, D., Neubig, G., & Berg-Kirkpatrick, T. (2019). Lagging inference networks and posterior collapse in variational autoencoders. arXiv preprint arXiv:1901.05534.
It would be interesting to compare at the conceptual level both heuristics.

R2: It seems that the authors do not use BatchNorm. Is it correct? This would be also interesting to discuss, because in the following paper:
Vahdat, A., & Kautz, J. (2020). Nvae: A deep hierarchical variational autoencoder. arXiv preprint arXiv:2007.03898.
the BatchNorm is indicated as an important component for achieving a deep VAE.

**Questions:**
Q1: What kind of prior was used in this paper?

Q2: Is BatchNorm indeed irrelevant for VAEs?

---

> ### Author Response · Authors · 2020-11-19
> **Thank you; updated**
>
> Dear Reviewer 1,
>
> Thank you for your careful review. We introduced a line in the body of the text explaining the prior (previously the information was only available in a caption). The prior is a diagonal Gaussian distribution, similar to Maaloe et al, 2019.
>
> With regard to batch normalization, we did not have time to fully study it, as Vahdat & Kautz (2020) was released toward the end of our experiments. Our initial experiments showed no gains from incorporating it, but our network also had too many differences with respect to Vahdat & Kautz to make any general statements about its necessity. We believe further investigating this would be valuable for future work.
>
> Thank you for the reference to Lagging inference networks, and thank you for your comments, as they will improve the clarity of our work.

---

> > ### Comment · AnonReviewer1 · 2020-11-23
> > **After the rebuttal**
> >
> > Dear authors,
> >
> > I would like to thank you for your comments. In my opinion, the paper is very important for our field, because it shows that we can achieve amazing scores for VAEs. I would even say that this paper is a landmark for us, and now we should figure out whether we can do better (e.g., smaller architectures, better bpds) and whether we really understand what is going on in VAEs (e.g., what is the essential component of).
> >
> > I am completely satisfied with the rebuttal, thus, I keep my original score, namely, 8.
> >
> > Best.

---

### Official Review · AnonReviewer2 · 2020-10-27
**Very good paper improving deep VAE performance beyond autoregressive models, ablation studies could further strengthen it**

**Rating:** 8
**Confidence:** 4

**Review:**

**summary**
the paper puts forward an idea that deep-enough VAE should perform at least as well as autoregressive models. Authors explore this in the context of image generation, and construct VAE model that is a generalisation of typical autoregressive architectures. They use several tricks to ensure stable training of very deep VAEs and show that final performance exceeds all autoregressive models. This experimentally supports their claim that very deep VAEs encompass autoregressive models.

**pros**
The idea of perceiving VAE architectures as strictly more powerful and potentially efficient is very appealing. Given the recent work on improving deep VAE training(like Vahdat & Kautz (2020)) this paper takes another step in this direction by effectively, as it seems from the text, removing the depth limitation for training such VAEs. The tricks used to stabilise training are pretty ad hoc, but their effectiveness, showed experimentally, is important in advancing the field.


**cons**
* The main criticism I have is around ablation studies that justify the proposed architecture choices and training stabilisation tricks, as well as comparison to other tricks in the literature (e.g. Vahdat & Kautz (2020)). Of course the positive result speaks for itself, but the paper would be even more convincing with some details on the exploration that led to the final model.



**questions**
* it would be good to clarify in the text how exactly sampled latent variables from lower layers are decoded into the images to produce Fig. 4: is the idea to pass those latents down the top-bottom path and just not add new latents in the node "+" within the topdown block?
* In Section 5.2.1, it is unclear why models with 32x32 and 1024x1024 resolutions have equal number of parameters: is this because ResNet blocks used at different resolutions share parameters?
* Did the authors experiment with methods of slowing down the training of the prior, other then stopping it for the first half of training? It seems that exponentially averaging prior parameters might be another way of doing it, although the exponent will become another hyperparameter.

**comments**
* Further investigating the relation between using NN interpolation in upsampling and having active latents in all layers would be very useful.
* I particularly enjoyed the perceptional shift that the paper advocates for, i.e. that VAE and autoregressive models are not competing approaches, but rather VAE is a more general one and it encompasses the latter.

---

> ### Author Response · Authors · 2020-11-19
> **Thank you; updated**
>
> Dear Reviewer 2,
>
> At your suggestion, we included ablation studies in the Appendix (please see our comment above for a summary). We were not able to directly compare the techniques of Vahdat & Kautz (2020), as the paper was released after most of our experiments were complete, and the code was not yet publicly available.
>
> For figure 4, we sample from the approx. posterior until the given resolution, then sample the rest from the prior at low temperature. This allows us to visualize what images are likely given some subset of latent variables. We added an explanation of this to the caption.
>
> The parameter counts are only approximately similar for these networks, which we clarified. We adjust layer counts for FFHQ-1024 and also reduce the width of the final high resolution layers. We detail this in a new hyperparameters table in the Appendix.
>
> As we noted in the comment above, we actually found that prior warmup was not required; apologies for this error in our first submission.
>
> Thank you for all these comments, as we believe these changes will improve the clarity of our presentation.

---

### Official Review · AnonReviewer3 · 2020-10-28
**A good paper: a clear idea supported by a robust set of experiments**

**Rating:** 7
**Confidence:** 4

**Review:**

1. Summary
This paper shows that deep hierarchical VAEs can outperform state-of-the-art autoregressive models on images. The authors first argue that autoregressive models are special cases of hierarchical VAEs and that hierarchical VAEs are universal approximators. They introduce a simple top-down (LVAE) architecture that scales past 70 layers. Furthermore, the model can be trained without using freebits or KL annealing -- although additional tricks are required (gradient skipping and prior warmup). They demonstrate that likelihood performance is correlated with depth and report state-of-the-art performances on multiple image datasets.

2. a Strong Points
- the contribution is clear: the idea the hierarchical models can outperform autoregressive models is clearly stated and supported by a short theoretical section and relevant experiments (likelihood vs. depth + benchmark)
- the proposed model is simpler than existing methods and is trained using a simpler objective (no freebits/KL-warmup):
- the authors demonstrate state-of-the-art likelihood on multiple datasets: the Very Deep VAE indeed outperforms large autoregressive models
- the authors demonstrate that the method is scalable: proof of concept on FFHQ 256 and 1024.

2. b Weak Points
- the part on parallel generation is unclear: are you referring to each of the latents for one layer being sampled independently? (i.e. $q(\mathbf{z}^l | \mathbf{z}^{l-1}, \mathbf{x}) = \prod_d q(z^l_d | \mathbf{z}^{l-1}, \mathbf{x})$)
- section 5.2.1 is unclear: "unlike autoregressive models, this scaling does not require greater training resources". In my opinion, using larger images does requires greater training resources because of the increased image definition.
- Although the code is provided, the experimental protocol is not well described in the paper (learning rate, number of epochs, hidden size, ...)

3. Recommendation
I recommend accepting this paper.

4. Recommendation Arguments
This paper presents a simple, yet solid, story. The empirical results strongly support that "very deep VAEs can outperform autoregressive models on images" and the authors introduced a minimal architecture allowing to do so.

5. Questions to the Author
Please elaborate on the weak points.

6. Feedback
Figure 2: the sentence "latent variables are observed variables" is odd. Latent variables are unobserved by definition.

---

> ### Author Response · Authors · 2020-11-19
> **Thank you; updated**
>
> Dear Reviewer 3,
>
> Thank you for your careful review. We would like to respond to the issues you raised.
>
> 1) Yes, that is what we meant by parallel generation. We have updated the text to explicitly state that.
> 2) We meant that a similarly sized network could be trained using a similar number of GPUs; you are correct that the higher resolution layers will technically consume more resources. We clarified this in the text.
> 3) We added a table to the Appendix containing the most important experimental hyperparameters.
> 4) We modified our statement to read: "Latent variables are identical copies of observed variables," to be more precise.
>
> These revisions improve the clarity of our work; thank you again for pointing them out.

---

> > ### Comment · AnonReviewer3 · 2020-11-23
> > **Thank you, this improves the quality of the paper**
> >
> > The paper has improved with the latest changes and this allows me to be more confident in recommending your work.
> >
> > Last detail:
> > 4. "Latent variables are identical to observed variables" and "Latent variables allow for parallel generation" remain quite unclear for the reader (how is it identical? how does it allow for parallel generation?). Maybe both text and figure could be updated to improve clarity.

---

### Public Comment · ~Christopher_Beckham1 · 2020-11-10
**Output distribution**

Hi there, great paper, it's good to see VAEs making somewhat of a comeback. It's nice that you provided some code too (and it was relatively straightforward to run).

I saw however that  you're actually using the discretised mixture of logistics (DMoL) for your output distribution p(x|z), rather than Gaussian (squared error). I don't know if this is a big confounding variable or not with regard to your table of results, but I think you should make it clear when this distribution is being used. I'd be curious to know for instance if this makes a big difference to the ELBO and FID. It wasn't obvious to me whether it made a big difference eye-balling samples between the two distributions, and I can say that at the very least that enabling this loss made training much slower. It may have even been the cause for some NaNs. That is probably due to an inefficient and/or unstable implementation, but it's worth knowing when training potentially very expensive models.

Thanks again.

---

> ### Author Response · Authors · 2020-11-19
> **DMOL is standard**
>
> Hi Christopher,
>
> Thanks for your comment -- we are happy you were able to run the code. With regard to the output distribution, all competitive non-autoregressive models (Flow++, BIVA, NVAE) also use the DMOL layer, as does the main autoregressive model we compare against (the PixelCNN++). Most other autoregressive models use Categorical distributions, which were shown to perform equivalently to the DMOL loss in the Image Transformer work (Parmar et al, 2018). Thus, we believe that DMOL is an apples-to-apples comparison with other work.
>
> With that said, model performance (both in ELBO and FID) will certainly be affected by using the mean squared error instead of a Gaussian output distribution[1], and between a Gaussian distribution and the DMOL. A detailed study of that is outside of the scope of our work, but we wish you luck!
>
> [1] Bin Dai and David Wipf. Diagnosing and enhancing vae models. arXiv preprint arXiv:1903.05789, 2019.

---

### Author Response · Authors · 2020-11-19
**Ablation studies added to Appendix**

We thank the reviewers for their detailed and careful feedback. One concern among multiple reviewers was a lack of ablation studies for the architectural components we introduced. We added these in the Appendix of the revision, and highlight the results here:

- Figure 5 shows the relationship between architecture components and posterior collapse. Residual connections greatly mitigate posterior collapse, and nearest-neighbor upsampling further ameliorates posterior collapse that can occur with convolutional upsampling.
- Table 3 shows that the residual scaling leads to fewer unstable updates across network sizes, and results in lower loss for networks with more than 30 layers.
- Figure 6 demonstrates the effect of gradient skipping, and indicates where networks would have diverged if updates were not skipped.

We furthermore found in the process of fixing a bug in our code that the prior warmup was not necessary for training. We removed references to it in the text, and uploaded the version of the code that does not use it. The bugfix also resulted in slightly improved performance on our benchmarks. The public code release will include pretrained models that can be evaluated to confirm the results we report.

We respond to individual reviewer questions in more detail below.

---

### Decision · Program_Chairs · 2021-01-07
**Final Decision**

**Decision:**

Accept (Spotlight)

**Comment:**

The paper posits that VAEs, if made sufficiently deep, are able to implement autoregressive models, and could possibly outperform them. Experimentally, the authors attempt make VAEs sufficiently deep so that they are able to outperform autoregressive models on image generation. The authors use a variety of tricks to scale the depth of the model to up to 78 stochastic layers, and achieve SOTA, or near-SOTA NLLs on a number of datasets. Furthermore, in comparison to other models (in particular the recently proposed Nouveau VAE), the models achieve these scores using far fewer parameters.

Although the tricks are a bit ad-hoc and the novelty is a bit weak, the experimental results are quite strong and would be of interest to anyone working on VAE research. Moreover, one of the weakness of the paper, a lack of ablations, was addressed during the rebuttal. All reviewers believed that the paper should be accepted, and I see nothing in the paper or the reviews to suggest otherwise.